# A novel *in vitro* model to study prolonged *Pseudomonas aeruginosa* infection in the cystic fibrosis bronchial epithelium

Meghan J. Hirsch[1,2], Emily M. Hughes[1,2], Molly M. Easter[1,2], Seth E. Bollenbecker[1], Patrick H. Howze IV[1], Susan E. Birket[1,2], Jarrod W. Barnes[1], Megan R. Kiedrowski[1,2], Stefanie Krick[1,2]*

**1** Division of Pulmonary, Allergy and Critical Care Medicine, Department of Medicine, The University of Alabama at Birmingham, Birmingham, AL, United States of America, **2** Gregory Fleming James Cystic Fibrosis Center, The University of Alabama at Birmingham, Birmingham, AL, United States of America

* skrick@uabmc.edu

**Data Availability Statement:** All relevant data are within the paper and its Supporting Information files.

## Abstract

*Pseudomonas aeruginosa* (PA) is known to chronically infect airways of people with cystic fibrosis (CF) by early adulthood. PA infections can lead to increased airway inflammation and lung tissue damage, ultimately contributing to decreased lung function and quality of life. Existing models of PA infection *in vitro* commonly utilize 1–6-hour time courses. However, these relatively early time points may not encompass downstream airway cell signaling in response to the chronic PA infections observed in people with cystic fibrosis. To fill this gap in knowledge, the aim of this study was to establish an *in vitro* model that allows for PA infection of CF bronchial epithelial cells, cultured at the air liquid interface, for 24 hours. Our model shows with an inoculum of 2 x 10² CFUs of PA for 24 hours pro-inflammatory markers such as interleukin 6 and interleukin 8 are upregulated with little decrease in CF bronchial epithelial cell survival or monolayer confluency. Additionally, immunoblotting for phosphorylated phospholipase C gamma, a well-known downstream protein of fibroblast growth factor receptor signaling, showed significantly elevated levels after 24 hours with PA infection that were not seen at earlier timepoints. Finally, inhibition of phospholipase C shows significant downregulation of interleukin 8. Our data suggest that this newly developed *in vitro* "prolonged PA infection model" recapitulates the elevated inflammatory markers observed in CF, without compromising cell survival. This extended period of PA growth on CF bronchial epithelial cells will have impact on further studies of cell signaling and microbiological studies that were not possible in previous models using shorter PA exposures.

## Introduction

Cystic fibrosis (CF) is an autosomal recessive disorder whose gene defect leads to dysfunction in the cystic fibrosis conductance regulator (CFTR) protein [1–4]. This protein defect gives rise to chloride ion dysregulation and thick mucus in epithelial tissue, including that of the gastrointestinal tract and the lungs, providing an optimal environment for opportunistic

**Funding:** This work was supported by the National Institute of Health (R01HL160911 S. K.) and the Cystic Fibrosis Foundation (KIEDRO18F5 and ROWE21R3 M.K.). The funders had no role in study design, data collection and analysis, decision to publish, or preparation of the manuscript.

**Competing interests:** The authors have declared that no competing interests exist.

bacterium such as *Pseudomonas aeruginosa* (PA) [2–5]. According to the Cystic Fibrosis Foundation, prevalence of chronic infection with PA in people with cystic fibrosis (pwCF) increases with age, starting between 25–34 years of age [6]. PA is a Gram-negative opportunistic pathogen that is well-known to cause debilitating infections in pwCF due to its ability to form biofilms, antibiotic resistance, and secretion of virulence factors [5, 7]. In addition, PA is the most common bacterial species to infect pwCF in adulthood [4, 8–12]. Despite these challenges, pwCF are living longer due to medical advances such as better hygiene practices, management therapies, and highly effective modulator therapies [13] that correct cystic fibrosis transmembrane conductance regulator (CFTR) production or function [3, 4, 14]. Unfortunately, chronic infection with PA is still a widespread problem for these people as although PA has been found to decrease initially among pwCF on HEMT, studies have found a resurgence occurs within a year or two from onset of the therapy [15–17]. This suggests that PA is not eradicated by HEMT and is still, therefore, a serious problem in pwCF [15–17] as PA can exacerbate chronic inflammation, leading to further lung tissue damage, decreased lung function, and ultimately respiratory failure [3, 4, 18, 19]. According to the US and European CF Foundation Respiratory Reports, respiratory failure continues to be the leading cause of death in pwCF [8, 9].

PA isolates from pwCF can have mucoid or non-mucoid phenotypes [3]. Non-mucoid phenotypes are more commonly associated with early infection and are typically motile with flagella and high levels of toxin and pyocyanin production [20]. Conversely, mucoid phenotypes produce more alginate [5, 20] and are associated with increased morbidity in pwCF [3, 20]. Additionally, PA produces biofilms composed of exopolysaccharides, proteins, lipids, and cytosolic proteins [20–22]. These biofilms are known to be prevalent in chronic PA infection in pwCF and make antibiotic therapy less effective [3, 22]. PA infections have been extensively studied from a microbiological perspective, and PA has been observed to develop biofilms by 6 hours (h) post-infection in co-culture with polarized CF airway epithelial cells [22]. However, this timepoint may not be long enough to investigate cellular signaling mechanisms that are affected by PA within the lung epithelium. Many downstream cellular mechanisms take 24–72 h to be detectable by modern assays due to the time it takes for transcription and translation to occur following activation by a stimulus [23–25]. Currently described *in vitro* infection models for PA lead to significant cell death by 8–12 h of infection [22].

To fill this knowledge gap, our lab has developed an *in vitro* method to infect human CF bronchial epithelial cells (CF-BECs), cultured and differentiated at the air liquid interface, with mucoid and non-mucoid PA for 24 h. We achieved this without significant cell death while still recapitulating the inflammatory response known to play a crucial role in chronic PA infections in pwCF.

## Materials and methods

### Cell culture and stimulation

Immortalized human bronchial epithelial cells (CFBE41o-) that are homozygous for the Δ508 CFTR mutation were used for these experiments [26, 27]. All CFBE41o- cells were cultured and maintained in Minimum Essential Media (MEM) with the addition of 10% fetal bovine serum, 0.5% Pen-Strep, 1% L-glutamine, and 0.2% Plasmocin at 37C with 5% $CO_2$. When flasks were 90% confluent, cells were seeded onto Vitrogen Plating Media (VPM) coated 12 mm Transwell filters, grown for 2–4 days with apical media, and then polarized at the air-liquid interface for 10–14 days. One day prior to bacterial infection, cells were washed with clear MEM containing 5% L-glutamine to remove residual antibiotics, and cells were switched to feeding with antibiotic-free MEM containing 5% L-glutamine and 10% FBS, without

plasmocin and Pen-strep. Where indicated, phospholipase C gamma (PLCγ) inhibitor (U73122; 3 μM) was used as a pre-treatment for 1 h before infection with PA.

### *P. aeruginosa* strains and infection

The non-mucoid PA strain PAO1 and the mucoid clinical isolate PAM57-15 were used for infection studies. PAO1 and PAM57-15 carrying the constitutively expressing GFP vector, pSMC21 [28], or the tdTomato expression vector pMQ400 [29] were used for fluorescence microscopy studies. PA strains were cultured in 5 mL of LB broth overnight at 37˚C, 1 day prior to assay. The PA infection protocol was modified from the static co-culture biofilm assay method by Anderson, G.G., et al, 2008 in Infection and Immunity [27]. Briefly, the overnight cultures of PA were washed twice with MEM + 5% L-glutamine without phenol red. An OD600 was obtained, and the OD 0.5 was calculated and made up from the washed overnight stock. Using the OD 0.5, inoculums were made by diluting 1:100, 1:500, and 1:1000 then filter size was accounted for. This led to the following mean CFUs; PAO1 $1.25 \times 10^3$, $2.42 \times 10^2$, and $1.18 \times 10^2$ and for PAM57-15 $1.01 \times 10^3$, $1.97 \times 10^2$, $1.07 \times 10^2$. These CFUs were then added to the apical surface of the respective wells. The apical media was taken off the cells at the 1 hr timepoint and centrifuged for 3 mins at 8000 rpm to remove any unbound PA. The supernatant (225μL/90μL) was put back to the apical surface with the addition of 4% L-arginine (25 μL/10μL) as previously described [27]. CFBE41o- cells were further co-cultured with PA in 5% $CO_2$ at 37˚C for 5, 11 or 23 h. 0, 1, 6 and 24 h CFU's were plated and counted where applicable.

### Cellular cytotoxicity assay

Cellular survival was quantified by measuring lactate dehydrogenase (LDH) release into the basolateral media [30]. LDH release was determined by using CytoTox 96® Non- Radioactive Cytotoxicity Assay Kit (Promega). This assay was performed according to the manufacturer's protocol, and percent survival was determined by calculating the inverse of cytotoxicity relative to uninfected, multiplied by 100%.

### Microscopy and immunofluorescence staining

Cells were treated as above on 12 mm Transwell filters. When indicated, brightfield images were taken after the apical media was taken off. Images were taken at 10X magnification on a Nikon Eclipse Ts2 inverted microscope. For immunofluorescent microscopy, cells were fixed with 4% paraformaldehyde overnight at 4˚C. After at least 24 h in fixative, cells were permeabilized with 0.1% Triton on ice for 15 mins. Cells were then incubated with the Hoechst 33342, trihydrochloride, trihydrate nucleic acid stain (3:1000, ThermoFisher) concurrently for 30 min at room temperature with shaking. The filters were mounted on microscope slides with Prolong Gold antifade reagent (Invitrogen) and sealed with coverslips. After mounting, microscope slides were protected from light and stored at 4˚C until imaging. Z-stacks were obtained on a Nikon A1R confocal microscope in the UAB High Resolution Imaging Facility. All images are at 60X magnification and show maximum intensity projections.

### IL-8 detection assay

After inoculation with control media (clear MEM without serum) or PA for 1, 6, 12, or 24 h, interleukin (IL)-8 was quantified in the basolateral media prepared for IL-8 Human Uncoated ELISA (Invitrogen). ELISAs were performed according to the manufacturer's protocol. The human IL-8 ELISA had a sensitivity of 2–250 pg/mL.

## Quantitative real time PCR assays

After inoculation with control media (clear MEM without serum) or PA for 1, 6, 12, or 24 h, total RNA was isolated using GeneJET RNA Purification Kit (ThermoScientific). RNA concentration for each sample was assessed using a Nanodrop and cDNA was synthesized using a Maxima™ H Minus cDNA Synthesis Master Mix with dsDNase kit (ThermoFischer). RT-qPCR was performed on an Applied Biosystems StepOnePlus using TaqMan primers Interleukin-8 (Hs00174103_m1, CXCL8), Interleukin-6 (Hs00174131_m1, IL-6), and Interleukin 1-β (Hs01555410_m1, IL-1β) and reference gene glyceraldehyde 3-phosphate dehydrogenase (GAPDH). Fold change was calculated as described in a previous publication in our lab [31]. In brief, the average CT value of duplicates of each treatment was subtracted by the cytokine (IL-6, IL-8, or IL-1β probe by GAPDH for a ΔCT, then taking the ΔCT for the treatment subtracted by the ΔCT of the time respective control. This number indicates the ΔΔCT value which was used to determine fold change using the following formula: POWER(2, -ΔΔCT).

## Protein immunoblotting

All protein lysates were obtained from CF-BECs using radioimmunoprecipitation assay buffer with phosphatase inhibitor, phosphatase inhibitor cocktail II (RPI), and protease inhibitor, Roche cOmplete™ Protease Inhibitor Cocktail (Millipore Sigma). Protein concentrations were determined by Bradford Assay. Subsequently, 40 μg of protein was loaded for each well. Proteins were separated on 4–20% precast Ready Gels (BioRad) and transferred onto nitrocellulose membranes (Cytiva). For loading control total protein, one gel was stained after protein separation using GelCode™ Blue Stain Reagent (Thermo Scientific) for 1 h. The gel was washed with deionized water overnight and imaged using Amersham Imager 600 system (GE). Membranes were blocked with 5% Bovine Serum Albumin (BSA) in Tris-buffered saline (pH 7.4) with 0.05% Tween 20 (TBST) for 30 mins and incubated overnight with the following primary antibodies: rabbit total and phospho-anti PLCγ (Cell Signaling Technologies). After 3 washes with TBST, membranes were incubated with goat anti-rabbit peroxidase conjugated (Invitrogen) at 1:6,000 in TBST for 1 h. Positive signals were visualized by chemiluminescence on an Amersham Imager 600 system (GE). Images were acquired using Image Lab software (Bio-Rad). Densitometry was measured using ImageJ software (National Institutes of Health). Densitometry was quantified by dividing P-PLCγ by total PLCγ [32]. These data were then normalized by total protein from the GelCode™ Blue Stain Reagent image.

## Statistical analysis

Data were analyzed using GraphPad Prism 9 for Macintosh (GraphPad Software). Three-way Anova, two-way Anova, mixed analysis test, and Kruskal-Wallis test were performed, followed by Tukey's multiple comparisons or Dunn's *post hoc* test using a 95% confidence interval where indicated. Un-paired t-tests using a 95% confidence interval were performed where indicated. Data are expressed as means ± standard error of mean (SEM). Differences between groups were considered statistically significant if $P < 0.05$.

# Results

## 24 hour PA infection on CF-BECs is feasible in an *in vitro* model

To overcome the limitations with the 6 h PA infection model, we developed a new methodology to extend PA infection for 24 h without cell death. (S1 Fig). The viability of CF-BECs 24 h post-infection was assessed using brightfield imaging and LDH assays. The non-mucoid strain (PAO1) at 1 x $10^3$ CFUs showed decreased cell density of the CF-BECs seen by the areas where

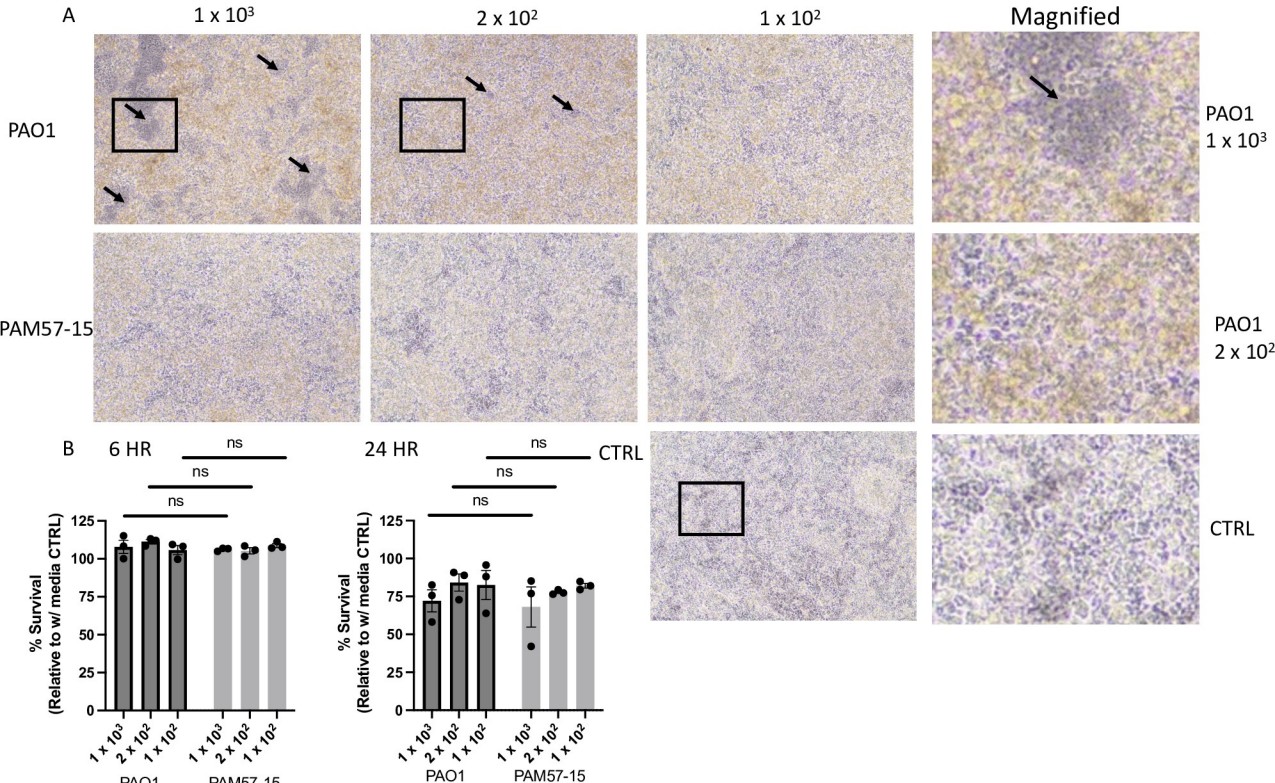

**Fig 1. CF-BECs demonstrate little change in viability after infection with PA at a dilution of $2 \times 10^2$ the OD 0.5 after 24 hours. A:** Representative bright field images of CF-BECs (CFBE41o-) on Air Liquid Interface (ALI) treated with PAO1 (non-mucoid) at mean CFUs of $1.25 \times 10^3$, $2.42 \times 10^2$, or $1.18 \times 10^2$ and PAM57-15 (mucoid) strains of PA at $1.01 \times 10^3$, $1.97 \times 10^2$, or $1.07 \times 10^2$ at 24 h post infection. Magnified images taken from PAO1 $1.25 \times 10^3$, $2.42 \times 10^2$, and the CTRL show differences in monolayer confluency. Black arrows indicate areas where the monolayer was broken. Yellow coloration indicates presence of PA. **B:** Graph showing % survival of cells at the 6 and 24 h timepoints measured by LDH assay (CytoTox 96Ⓡ Non-Radioactive Cytotoxicity Assay Kit -Promega). Statistical analysis was done using a two-way Anova and a mixed effects analysis respectively followed by Tukey's multiple comparisons test with a 95% confidence interval showing mean ± S.E.M. with *$P<0.05$.

the monolayer was broken or "cell free areas" compared to the $2 \times 10^2$ and $1 \times 10^2$ dilutions (Fig 1A, arrows pointing to "cell free areas"). This is seen in more detail in the magnified images of PAO1 at $1 \times 10^3$ and $2 \times 10^2$ compared to the control (Fig 1A). In contrast to the $1 \times 10^3$ dilution, the dilutions at $2 \times 10^2$ and $1 \times 10^2$ showed little to no cell density decreases qualitatively compared to the control (Fig 1A). In addition, the mucoid strain appeared to have less effect on confluency than the non-mucoid PA strain shown by little to no areas where the monolayer was broken or "cell free areas" (Fig 1A). According to the LDH assay, cells exposed to $1 \times 10^3$ of PA whether mucoid or non-mucoid in phenotype at 24 h showed an average of 70% survival compared to 100% survival in their 6 h infected counterparts (Fig 1B). The CF-BECs infected with $2 \times 10^2$ dilutions of PA showed better survival rates, above 75% (Fig 1B). Overall, the cell viability of the human bronchial epithelial cells after infection with PA at 24 h is higher than 75% with a dilution of $2 \times 10^2$ of the OD0.5 of PA, indicating that most of these cells survive extended PA infection *in vitro*.

## The bacterial burden of CF-BECs post-infection at 24 hours is increased compared to the 6 hour timepoint

To determine the bacterial burden at the 0, 1, 6, and 24 h timepoints, we plated CFUs of PA expressing a tdTomato plasmid, and subjected cells to confocal microscopy imaging of the

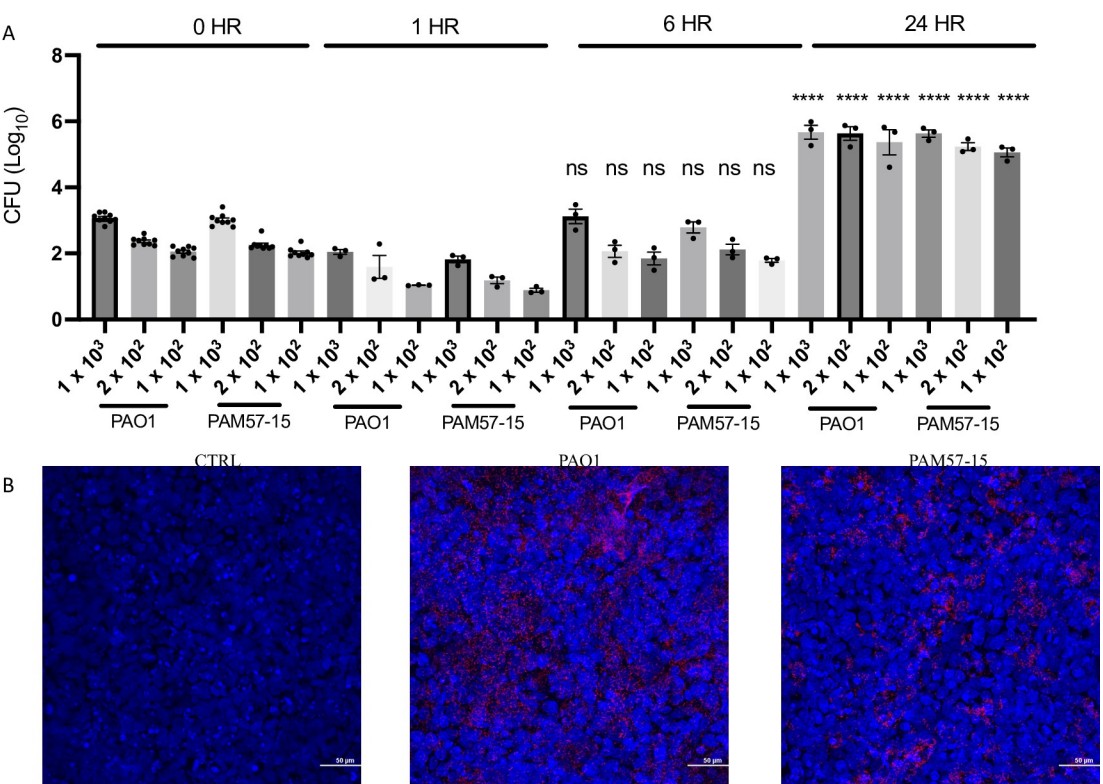

**Fig 2. CF-BECs infected with 2 x 10² CFU's of PA demonstrate significant bacterial burden at 24 hours without affecting monolayer confluency. A:** Graph illustrates Log₁₀ CFUs at 0, 1, 6, and 24 h for CF-BECs, cultured at the ALI and infected with PAO1 and PAM57-15 strains of PA. **B:** Representative 40X confocal microscopy images of CF-BECs infected with PAO1, and PAM57-15 infected with 2.42 x 10² and 1.97 x 10² mean CFUs respectively with plasmids fluorescently tagged with tdTomato after 24 h. Statistical analysis was done using a three-way Anova followed by Tukey's multiple comparisons test with a 95% confidence interval showing mean ± S.E.M. with ****$P<$0.0001, when compared to the 0 h respective controls.

epithelial cell nuclei (Hoechst staining) (Fig 2). The CFUs at the 0 h timepoint were plated for each timepoint and the mean CFUs for PAO1 were 1.25 x 10³, 2.42 x 10², and 1.18 x 10² respectively. The mean CFUs for PAM57-15 were 1.01 x 10³, 1.97 x 10², and 1.07 x 10². Little variability within the 0 h timepoint at each dilution supports that the infection dilution technique was precise and consistent (Fig 2A). The 1 h CFUs for all dilutions were lower than the 0 h, showing that only some of the PA adhered to the cells by the 1 h timepoint. By 6 h, however, the PA has increased almost to the 0 h CFUs after the 1 h wash as described in the methods. Interestingly, by 24 h, mean CFUs were 5.73 x 10⁵, 5.15 x 10⁵, and 3.89 x 10⁵ for PAO1 and 4.54 x 10⁵, 1.85 x 10⁵, and 1.24 x 10⁵ for PAM57-15 compared to the 6 h timepoint which were 1.67 x 10³, 1.36 x 10², and 8.4 x 10¹ for PAO1 and 6.94 x 10², 1.52 x 10², and 6.26 x 10¹ for PAM57-15 respectively. Overall, these data show a 1 x 10³ increase in bacterial burden from 6 to 24 h (Fig 2A). In addition, the confocal microscopy images showed presence of both PAO1and PAM57-15 at 2 x 10² (red) compared to the control at the 24 h timepoint (Fig 2B). Overall, there is an increased amount of PA burden on the CF-BECs at the 24 h timepoint compared to the 6 and 1 h timepoint.

## Pro-inflammatory markers are increased with prolonged PA infection

To determine whether this model elicits a pro-inflammatory response, similar to what has been shown previously using shorter infection times [18, 33, 34], we analyzed mRNA

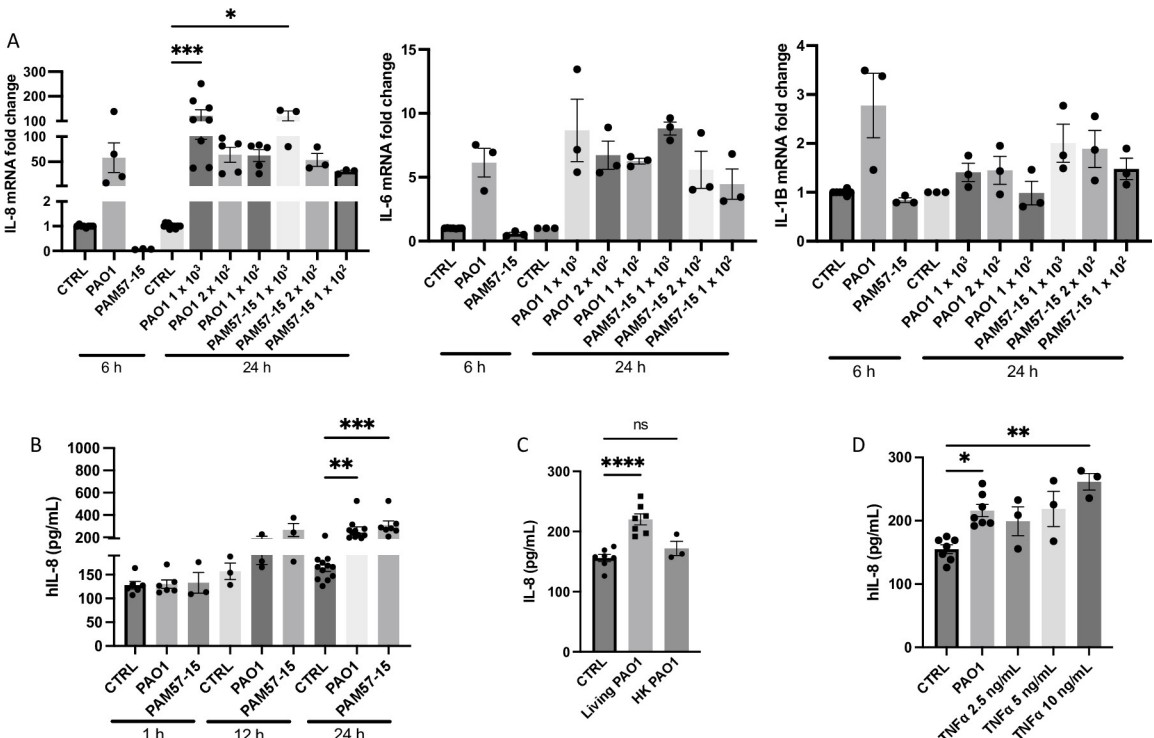

**Fig 3. PA infection increases IL-8, IL-6, and IL-1β mRNA levels and IL-8 protein levels in CF-BECs. A:** Fold change levels of CXCL8 (IL-8), IL-6, and IL-1β mRNA levels at the 6 and 24 h timepoints when treated with vehicle (CTRL), or PA strains (PAO1 and PAM57-15). **B:** Graph showing IL-8 protein levels from basolateral media of CF-BEC ALI cultures when treated with CTRL, PAO1, or PAM57-15 for 1, 12, or 24 h. **C:** Graph showing levels of IL-8 secretion into the basolateral media when CF-BECs are treated with heat killed (HK) or living PA at 24 h **D:** Diagram illustrating IL-8 secreted protein levels after treatment with TNFα (2.5, 5, and 10 ng/mL) compared to PAO1 infection. All 1, 6, and 12 h timepoints were treated with an OD 0.5 inoculum to mimic what has been used in literature unless otherwise described. 24 h treatments had 0 h CFUs of $2 \times 10^2$ unless otherwise described. Data was represented as fold change in mRNA expression with at least n = 3 independent experiments. Statistical analysis was done using Kruskal-Wallis test, followed by Dunn's *post hoc* test using a 95% confidence interval showing mean ± S.E.M. with *$P<0.05$, ** $P<0.01$, ***$P<0.005$, and ****$P<0.0001$, compared to time respective control (ctrl) group.

expression levels of IL-1β, IL-6 and CXCL8 (IL-8) along with protein levels of IL-8 in our model. We chose to further analyze IL-8, given its role as a potent neutrophil chemokine, which is well-known to be upregulated with PA infection in CF [33, 35, 36]. We first quantified transcript levels of CXCL8 (protein: IL-8) which was shown to be significantly upregulated at $1 \times 10^3$ and $2 \times 10^2$ of PAO1 or PAM57-15, averaging to around a 50-fold increase compared to the control at the 24 h timepoint (Fig 3A). We then quantified transcript levels for IL-6 and IL-1β. IL-6 showed around a 6-fold increased at the $1 \times 10^2$ dilutions compared to the uninfected control at the 24 h timepoint (Fig 3A). IL-1β levels showed a trending slight increase compared to the control with most dilutions at the 24 h timepoint (Fig 3A). Additionally, 24 h timepoint dilutions were at or above the level of IL-8 and IL-6 seen at the 6 h timepoint regardless of the PA strain (Fig 3A). Furthermore, protein levels of secreted IL-8 in the basolateral media using an ELISA also showed a significant increase at the 24 h timepoint compared to the control (Fig 3B). To determine whether viable PA was required for this effect, we used heat inactivated PA as a control for the inoculation of CF-BECs. Heat inactivated PA did not induce IL-8 secretion as the living PA did, indicating the importance of viable PA to exert the observed pro-inflammatory response (Fig 3C). To assess the effect size of the PA-induced inflammatory response, we compared prolonged PA inoculation to stimulation of CF-BECs with tumor

necrosis factor (TNF) α, which is known to stimulate IL-8 secretion in these cells [37]. As shown in Fig 3D, TNFα stimulated IL-8 production in a concentration-dependent manner, comparable to the effect elicited by the 24 h PA infection. In summary, our model shows elevated IL-1β, IL-6 and CXCL8 mRNA levels after 24 h infection with PA. In addition, secreted IL-8 is significantly elevated in the basolateral media from CF-BECs at 24 h, which is dependent on live PA.

## PA induces phosphorylation of phospholipase C (PLC)γ at 24 hours

To determine which potential signaling pathways are activated at the 24 h time point by PA, we probed for total and phosphorylated PLCγ via immunoblotting after infection with either mucoid or non-mucoid PA. At 1 h of infection, phospho-PLCγ was not upregulated compared to the control in the CF-BECs infected with non-mucoid PA, while there was a trend increase after 12 h the effect was even more pronounced by 24 h (Fig 4A and 4B). In the CF-BECs that were infected with mucoid PA, by 24 h there was a significant increase in phospho-PLCγ in the PA infected cells compared to its respective control (Fig 4C and 4D). PLCγ inhibitor, U-73122, was then used as a pre-treatment on CF-BECs to determine if blocking PLCγ would decrease IL-8 upregulation. With infection with PAM57-15 and U-73122 together for 24 h, IL-8 levels in the cell lysate significantly decreased compared to PAM57-15 only IL-8 levels (Fig 4E). In summary, PA infection led to phosphorylation of PLCγ after 24 hrs and inhibition of PLCγ lead to decreased levels of IL-8 production in the cell lysate.

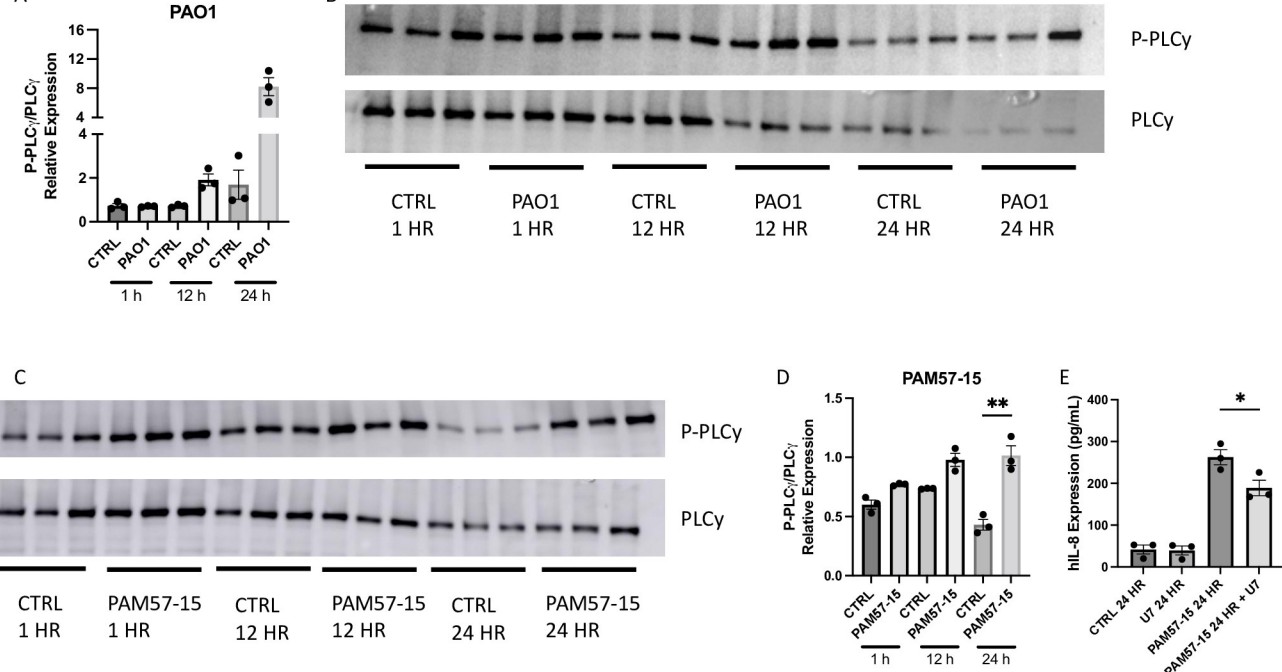

**Fig 4. P-PLCγ is upregulated at 24 h with PA infection compared to the uninfected control. A:** Graph showing quantification of P-PLCγ/PLCγ/Total for PAO1 infected and uninfected cells. **B:** Representative western blots for P-PLCγ and PLCγ of PAO1 infected and uninfected cells at 1,12, and 24 h. **C:** Representative western blots for P-PLCγ and PLCγ of PAM57-15 infected and uninfected cells at 1,12, 24 h. **D:** Graph showing quantification of P-PLCγ/PLCγ/Total for PAM57-15 infected and uninfected cells. **E:** Graph showing IL-8 levels in the cell lysate of CF-BECs treated with/without PAM57-15 and with/without 1 h pre-treatment with PLCy inhibitor (U-73122) at 3 μM. Western blot quantification was normalized to total protein using Coomassie staining. ELISA data was analyzed using an un-paired t-test. All 1 and 12 h timepoints were treated with an 0 h CFU count made from the OD 0.5 to mimic what has been used in literature unless otherwise described. 24 h treatments had 0 h CFUs of 2 x 10² unless otherwise described. Statistical analysis was done using Kruskal-Wallis test, followed by Dunn's *post hoc* test using a 95% confidence interval. All n = 3 independent experiments showing mean ± S.E.M. with *P<0.05 and **P<0.01, compared to time respective control (ctrl) group.

## Discussion

In this study, we are one of the first to develop a bacterial co-culture model using CF-BECs cultured at the ALI that extends the timeframe for PA infection *in vitro* past 6 to 24 h. This model recapitulates the increased levels of IL-8 and IL-6 elicited by PA infection, without sacrificing cell viability. In prior studies, most *in vitro* infections ended at 6 h post-inoculum [12, 38, 39]. This timepoint has become rather common for PA infection studies, not only for cell viability reasons but also due to the start of production of PA biofilms around this time, thus deeming this timepoint a chronic infection model [22]. Other studies used antibiotics such as gentamicin or tobramycin to prevent PA overgrowth and subsequent cell viability issues during PA *in vitro* infection [27, 40]. Moreau-Marquis et al. completed a time course experiment with PA and noticed decreased cell viability starting around 10 h post-infection, and therefore, the maximum time used for observing infection was 8 h [22].

Given these studies, one of the benefits of our novel 24 h *in vitro* infection model is the ability to avoid the use of antibiotics, therefore excluding their potential effects on PA and CF-BECs, allowing to elucidate the interactions between the bacteria and host. In addition, our model establishes the potential for further studies analyzing the effect of chronic PA infection on bronchial epithelial cellular signaling mechanisms. For instance, elucidating the mechanism by which phospholipase gamma is contributing to increases in pro-inflammatory marker IL-8 at this prolonged timepoint. Those results are impactful in furthering our understanding of the host/pathogen interactions occurring in chronic infections in CF patients.

Although this model does not elucidate the specific PA product(s) that may be responsible for downstream cellular effects, it may serve as a starting point for further investigation of these topics. While a difference in cell viability between the 6 and 24 h timepoints was observed, the difference seen with the $10^2$ 0 h timepoint compared to the $10^3$ of PA is encouraging at above 75% (Fig 2B). The ability to observe prolonged PA exposure with minimal cell death is a benefit that will facilitate further studies in this chronic model of PA infection in the CF bronchial epithelium at ALI.

In addition to furthering CF research, this model can be adapted for other pulmonary diseases where PA infection is commonly observed, including but not limited to chronic obstructive pulmonary disease (COPD) and non-CF bronchiectasis. Alternatively, this approach could also be used for other bacteria. In conclusion, this model affords for the ability to take a deeper look into the effects that PA has on downstream cellular mechanisms in the human bronchial epithelium that could not be achieved with the established infection timeline. Overall, extending *in vitro* infections with PA allows for the ability to discover more about the host-pathogen interactions in addition to testing novel therapeutic options using this model.

## Supporting information

**S1 Fig. Diagram of methodology of PA 24-hour infection in a CF bronchial epithelial cell** ***in vitro*** **model.** Created with BioRender.com.
(TIF)

**S2 Fig.**
(TIF)

**S3 Fig.**
(TIF)

**S4 Fig.**
(TIF)

**S5 Fig.**
(TIF)

**S6 Fig.**
(TIF)

## Author Contributions

**Conceptualization:** Meghan J. Hirsch, Megan R. Kiedrowski, Stefanie Krick.

**Data curation:** Meghan J. Hirsch, Emily M. Hughes, Molly M. Easter, Seth E. Bollenbecker, Patrick H. Howze IV.

**Formal analysis:** Meghan J. Hirsch, Molly M. Easter, Seth E. Bollenbecker, Susan E. Birket, Jarrod W. Barnes, Megan R. Kiedrowski, Stefanie Krick.

**Methodology:** Meghan J. Hirsch.

**Writing – original draft:** Meghan J. Hirsch, Megan R. Kiedrowski, Stefanie Krick.

**Writing – review & editing:** Meghan J. Hirsch, Emily M. Hughes, Molly M. Easter, Seth E. Bollenbecker, Patrick H. Howze IV, Susan E. Birket, Jarrod W. Barnes, Megan R. Kiedrowski, Stefanie Krick.

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
