## [Decision Letter · Decision Letter 0]

17 Feb 2023

PONE-D-22-33243A novel * in vitro * model to study prolonged * Pseudomonas aeruginosa * infection in the cystic fibrosis bronchial epitheliumPLOS ONE

Dear Dr. Stefanie,

Thank you for submitting your manuscript to PLOS ONE. After careful consideration, we feel that it has merit but does not fully meet PLOS ONE’s publication criteria as it currently stands. Therefore, we invite you to submit a revised version of the manuscript that addresses the points raised during the review process.

We look forward to receiving your revised manuscript.

Kind regards,

Harish Chandra, PhD

Academic Editor

PLOS ONE

Journal Requirements:

Journal Requirements:

Reviewers' comments:

Reviewer's Responses to Questions

**Comments to the Author**

1. Is the manuscript technically sound, and do the data support the conclusions?

Reviewer #1: Partly

Reviewer #2: Yes

2. Has the statistical analysis been performed appropriately and rigorously? 

Reviewer #1: Yes

Reviewer #2: Yes

3. Have the authors made all data underlying the findings in their manuscript fully available?

Reviewer #1: No

Reviewer #2: No

4. Is the manuscript presented in an intelligible fashion and written in standard English?

Reviewer #1: Yes

Reviewer #2: Yes

5. Review Comments to the Author

Reviewer #1: The manuscript entitled “A novel in vitro model to study prolonged Pseudomonas aeruginosa infection in the cystic fibrosis bronchial epithelium” by Meghan et al. to explore novel invitro infection model is very interesting. The manuscript is well written however, the results lack crucial data points and are not well represented. A comparison with 1-6 hours infection model needs to be shown in all data points.

Therefore, I feel the current version is not suitable for publication in PlosOne until the following major concerns are addressed.

Figure 1: Bright field image of the cells is not clear.

Figure2: CFUs for both inoculum used and endpoint should be converted to Log CFUs for better representation. Further, the inoculum CFUs and endpoint CFUs may be plotted together for better comparision.

The authors should also include 0 hours CFUs Post infection after washing of the unbound bacteria.

Figure 3A: mRNA fold change for Il-8, IL-6, IL-1B, for 6 HR data is shown only for PA01. At what dilution was this data obtained? Why the authors did not include other dilutions (1:100, 1: 500, 1:1000) and the mucoid infections as shown for the 24 HR infection? Since the manuscript compares the 6 hours infection model therefore, authors must replicate the 6 HR data points as shown for 24 HR model.

Similar problem is associated with Fig 3B, C and D. Authors should include all the data points.

Fig 3C, the control cells have high levels of Il-8 while we do not see significant mRNA increase?

Fig 4: The western blot do not show the loading control band (GAPDH or Actin). The phosphorylated and the total proteins should be compared with the loading control in each well for densitometric analysis.

It is difficult to conclude that the there is increase in phosphorylation. In fact, it looks other way round if you compare it with the untreated control data points!

Fig 4E is not clear!

Reviewer #2: Hirsch et al. has provided a well-written manuscript on Pseudomonas aeruginosa in people with CF with nice translational potential with their in vitro model. Their science is sound, and their research contribution is valuable.

I’ve attached some minor corrections, below. Other than that, I would highlight the aim of the study and the impact in the discussion.

Unfortunately, I can’t seem to see the figures so I cannot comment on that.

Well don, Hirsch et al!

L36: comma after tissue

L37: comma after lungs

L39: remove space between 25-34

L40: capital G for Gram

L47,49: ‘therapy’ following HEMT seems redundant?

L80: phospholipase C gamma (PLCy)

L92: remove ‘-’ (i.e., 1 h)

L107,112: 4 °C

L113: 60× magnification (use symbol not the letter x)

L130-131: Roche cOmplete™ Protease Inhibitor Cocktail

L132: elsewhere you’ve not cited location with company, i.e. Millipore Sigma (L131), L124: ThermoFischer. Please keep it consistent throughout.

L146: were (?) performed

L147: standard error of mean (SEM)

L148: P is always capitalised and italicised

L150: use hour in subheading

L166: Air Liquid Interface (ALI)

L170-171: use hours, hour in subheading

L187: tdTomato

L216: italicise ‘post hoc’

L244: 6 to 24

Throughout: hrs is an incorrect abbreviation of hours. Instead of hr / hrs, use h.

6. PLOS authors have the option to publish the peer review history of their article (what does this mean?). If published, this will include your full peer review and any attached files.

Reviewer #1: No

Reviewer #2: No

---

## [Author Response · Author response to Decision Letter 0]

14 Apr 2023

Please see Response to Reviewers document for a point-by-point response to each comment. Thank you.

---

## [Editor Report · Decision Letter 1]

26 Apr 2023

PONE-D-22-33243R1A novel * in vitro * model to study prolonged * Pseudomonas aeruginosa * infection in the cystic fibrosis bronchial epitheliumPLOS ONE

Dear Dr. Krick,

Thank you for submitting your manuscript to PLOS ONE. After careful consideration, we feel that it has merit but does not fully meet PLOS ONE’s publication criteria as it currently stands. Therefore, we invite you to submit a revised version of the manuscript that addresses the points raised during the review process. Please submit your revised manuscript by Jun 10 2023 11:59PM. If you will need more time than this to complete your revisions, please reply to this message or contact the journal office at plosone@plos.org. Please include the following items when submitting your revised manuscript:A rebuttal letter that responds to each point raised by the academic editor and reviewer(s). You should upload this letter as a separate file labeled 'Response to Reviewers'.A marked-up copy of your manuscript that highlights changes made to the original version. You should upload this as a separate file labeled 'Revised Manuscript with Track Changes'.An unmarked version of your revised paper without tracked changes. You should upload this as a separate file labeled 'Manuscript'.

We look forward to receiving your revised manuscript.

Kind regards,

Harish Chandra, PhD

Academic Editor

PLOS ONE

Additional Editor Comments:

The manuscript entitled “A novel in vitro model to study prolonged

Pseudomonas aeruginosa infection in the cystic fibrosis bronchial epithelium” by Meghan

et al. to explore novel invitro infection model was revised. However, there are still some concerns that needs to be resolved.

Therefore, I feel the current version is not suitable for publication in PlosOne until the

following major concerns are addressed.

Line 130: gene glyceraldehyde 3-phosphate dehydrogenase (GAPDH) was used as reference in QRT PCR, while the authors dispute the use of GAPDH and B actin as controls in the western blot. Please use another internal control for fold change calculations. How was the fold change calculated? Please describe in methods.

Did the authors quantify the total proteins in the lysates before loading? The authors should use equal amount of proteins in each well for normalization. How the results were compared in densitometer analysis?

The authors used various dilutions for infections (1: 100, 1:500, 1:1000),. For better comparison, the authors should have optimized it with real numbers of CFUs.

Figures provided are very blurred and not clear. Therefore, not suitable for publication at this stage.

Please provide clear Images in fig 1A and 2B.

All figures provided should be clear and distinct. Error bar font sizes are too big and placed irregularly.

Thanks

---

## [Author Response · Author response to Decision Letter 1]

8 May 2023

The methods have been updated for RT-qPCR and for the Immunoblots. Please see Response to Editor for a point-by-point explanation regarding the changes that were made.

Additionally, the Funding Section has been removed from the manuscript as requested.

Thank you.

---

## [Editor Report · Decision Letter 2]

18 May 2023

PONE-D-22-33243R2A novel * in vitro * model to study prolonged * Pseudomonas aeruginosa * infection in the cystic fibrosis bronchial epitheliumPLOS ONE

Dear Dr. Krick,

Thank you for submitting your revised manuscript to PLOS ONE. After careful consideration, we feel that it has merit  and is very close to publication. Therefore, we invite you to submit a revised version of the manuscript that addresses the points raised during the review process.

We look forward to receiving your revised manuscript.

Kind regards,

Harish Chandra, PhD

Academic Editor

PLOS ONE

Journal Requirements:

Additional Editor Comments:

Minor Comments:

Description of Fig 2 legends is superficial. Fig 2A and its description is not clear. Please improve the bar diagram and the error bars specifically the round filled dots (Reduce the size).

Is the inoculum same to infect at different time points (1 hr, 6h and 24 hr)?

If yes then it should not be repeated for every time points instead it should be mentioned as zero time point (zero hours, 1 hours, 6 hours 24 hours). The term endpoints should be then removed!

The inoculum and the endpoint nomenclature is not clear in the experimental description.

(Dilutions of the initial inoculum may be mention as 102, 0.5 x 103 , 103 etc.

All figure legends should be described in detail including the statistical analysis.

---

## [Author Response · Author response to Decision Letter 2]

2 Jun 2023

Thank you for your comments on our manuscript. The inoculum/endpoint nomenclature has been revised throughout the manuscript and figures. Specific changes can be found in the track changes manuscript version. All references have been reviewed and none have been retracted. A further point-by-point explanation of the changes made per your comments can be found in the 06-01-2023 Responds to Editor document. Additionally, all response to reviewers/editors have been removed outside of the most current one.

---

## [Editor Report · Decision Letter 3]

19 Jun 2023

A novel * in vitro * model to study prolonged * Pseudomonas aeruginosa * infection in the cystic fibrosis bronchial epithelium

PONE-D-22-33243R3

Dear Dr. Stefanie Krick,

Thanks you for submiiting your revised manuscript. After careful reviewing, we found that the all the major concerns raised during reviewe process have been addressed and the manuscript has been improved significantly. Therefore, we’re pleased to inform you that your manuscript has been judged scientifically suitable for publication and will be formally accepted for publication once it meets all outstanding technical requirements.

Kind regards,

Harish Chandra, PhD

Academic Editor

PLOS ONE

---

## [Editor Report · Acceptance letter]

2 Jul 2023

PONE-D-22-33243R3 

A novel *in vitro* model to study prolonged *Pseudomonas aeruginosa* infection in the cystic fibrosis bronchial epithelium 

Dear Dr. Krick:

I'm pleased to inform you that your manuscript has been deemed suitable for publication in PLOS ONE. Congratulations! Your manuscript is now with our production department. 

Kind regards, 

on behalf of

Dr. Harish Chandra 

Academic Editor

PLOS ONE